# Vincristine-Induced Neuropathy in Patients Diagnosed with Solid and Hematological Malignancies: The Role of Dose Rounding

**DOI:** 10.3390/jcm12175662

**Published:** 2023-08-31

**Authors:** Abdulrahman M. Alwhaibi, Ali A. Alshamrani, Miteb A. Alenazi, Shroog F. Altwalah, Nouf N. Alameel, Noura N. Aljabali, Sara B. Alghamdi, Abdulwahab I. Bineid, Monira Alwhaibi, Mohamed N. Al Arifi

**Affiliations:** 1Department of Clinical Pharmacy, College of Pharmacy, King Saud University, Riyadh 11451, Saudi Arabia; nouf.n.alameel@gmail.com (N.N.A.); noura.n.aljabali@gmail.com (N.N.A.); sarabalghamdi1@gmail.com (S.B.A.); awahab.i.bineid@gmail.com (A.I.B.); malwhaibi@ksu.edu.sa (M.A.); malarifi@ksu.edu.sa (M.N.A.A.); 2Department of Pharmacology and Toxicology, College of Pharmacy, King Saud University, Riyadh 11451, Saudi Arabia; aaalshamrani@ksu.edu.sa; 3Pharmacy Department, Medical City (KSUMC), King Saud University, Riyadh 11451, Saudi Arabia; shroogaltw@hotmail.com

**Keywords:** vincristine, neuropathy, dose rounding, cancer, solid tumors, hematologic malignancies

## Abstract

Background: Vincristine is a vital constituent of chemotherapeutic regimens. Vincristine-induced neuropathy is a challenging adverse effect that impacts quality of life and treatment course. The dose rounding of chemotherapies is a strategy that is commonly used in clinical practice. Nevertheless, the frequency of developed neuropathy in vincristine first-time users and the potential association with dose rounding remains elusive. Methods: A retrospective analysis was conducted on patients administered vincristine for the first time between 2016 and 2022 using the King Saud University Medical City (KSUMC) database. Patients were stratified into pediatric and adult groups. Neuropathy frequency, its association with demographic and clinical parameters, and the Impact of dose rounding were assessed using SPSS software version 28. Results: Approximately 34.6% of patients were diagnosed with neuropathy after vincristine administration. Autonomic neuropathy was common among affected adults and pediatric patients (55.1% and 56.1%, respectively), while cranial neuropathy was more frequent in pediatric patients. Higher BSA (*p* = 0.038) and Scr (*p* = 0.044) in the pediatric group, the presence of respiratory comorbidities (*p* = 0.044), and the use of azole antifungals (*p* < 0.001) in the adult group were significantly associated with neuropathy episodes. The rounding-up of vincristine doses was significantly associated with increased neuropathy occurrence (*p* < 0.001), while dose rounding-down was significantly associated with a decrease in neuropathy in both groups of patients (*p* < 0.001). Conclusions: Our findings demonstrate that autonomic neuropathy is the most common vincristine-related neuropathy, regardless of the patient’s age. Dose rounding is a significant determinant of vincristine-induced neuropathy in both groups. Further studies are needed to evaluate the variables that exacerbate or prevent neuropathy associated with the first-time use of vincristine.

## 1. Introduction

Vincristine is a chemotherapeutic agent belonging to the group of vinca alkaloids, which has been frequently used in cancer treatment regimens since its approval for treating acute lymphoblastic leukemia (ALL) in the early 1960s [1]. This class of antineoplastic drugs has been extensively incorporated in multi-agent treatment protocols for several solid tumors, and hematologic and lymphatic malignancies [2,3]. Vincristine exerts antitumor activity by inhibiting the polymerization of tubulin, thus preventing mitotic spindle assembly and causing mitosis arrest at the metaphase, hence being called anti-mitotic drugs [4]. Similar to other antineoplastic agents, intolerable side effects, such as vincristine-induced neuropathy, commonly impact the patients’ quality of life (QoL) and consequently influence treatment continuation and outcomes [5].

Neuropathy, particularly autonomic and peripheral sensory–motor neuropathy, is a dose-limiting and dose-dependent adverse effect frequently reported in cancer patients treated with vincristine [6,7]. Neuropathy can be categorized as acute or chronic and usually appears within a couple of weeks of treatment initiation [6]. Vincristine-induced peripheral neuropathy (VIPN) is the most common type and often manifests as neurotic pain, paresthesia, muscle weakness, wrist and foot drop, and areflexia [7,8,9,10]. Transient blindness, ptosis, diplopia, jaw pain, facial palsy, sensorineural hearing loss, and laryngeal nerve paresis can occur; however, they are symptoms of cranial neuropathy [11]. Autonomic neuropathy occurs less frequently and can manifest as urinary retention and constipation [12]. It is worth mentioning that some neuropathic symptoms remain for a long period of time, even after completing a treatment regimen [3]. Moreover, their severity has been strongly linked with other factors, including patient characteristics, administration setup, pharmacokinetics, and genetic predispositions [13]. Additionally, vincristine is primarily metabolized in the liver by cytochrome P450 (CYP) 3A enzymes; thus, the concomitant administration of competitive CYP3A enzyme inhibitors, such as azole antifungals, has been linked to more adverse effects, including neuropathy, secondary to spikes in its plasma concentrations [14,15,16,17]. Therefore, dosing adjustments and continuous monitoring are of high importance to maintain vincristine concentrations within the therapeutic range and minimize the risk of neuropathies in adult and pediatric patients.

The dose rounding of biological and cytotoxic agents within 5–10% of the ordered dose to eliminate the use of the next vial is a commonly implemented clinical practice to minimize waste and cut down healthcare expenditures without losing its clinical effectiveness [18,19,20]. However, whether this practice would impact the neuropathy episodes associated with vincristine remains unknown. In our study, we aim to investigate the frequency of vincristine-associated neuropathy and determine whether dose rounding influences their occurrence among cancer patients for the first administration of vincristine-based chemotherapy in Saudi Arabia.

## 2. Methods

### 2.1. Subjects and Database

This study was a retrospective audit of patients admitted to the oncology center at King Saud University Medical City (KSUMC) between 2016 and 2022 and received, for the first time, at least one dose of a vincristine-based chemotherapy regimen (Figure 1). The Institutional Review Board at the KSUMC reviewed and approved the project (IRB Project No. E-21-6204). Electronic medical records were retrieved from the KSUMC database and reviewed for demographic data, clinical diagnoses, clinical parameters, and comorbidities before vincristine initiation. The date of vincristine administration for each patient (pediatric and adult) was collected, and patients were tracked after that until the end of 2022 or if they completed their chemotherapy, whichever came first. Protocols of vincristine-based chemotherapy were noted and the doses of vincristine per protocol were documented. The method of vincristine administration; total number and amount of actual doses received; total number and amount of doses calculated based on the body surface area (BSA) for each patient; type of dose rounding of each dose (if available); number of rounded, non-rounded, and missing doses; symptoms of neuropathy; types of neuropathy; interventions performed to manage neuropathy; time elapsed since vincristine initiation until neuropathy diagnosis (if reported); time to recover from neuropathy; and use of azole antifungal medications while receiving vincristine treatment were documented.

### 2.2. Study Design and Setting

To assess the impact of dose rounding on neuropathy, all vincristine doses administered to every patient were calculated based on their BSA, measured just before their scheduled doses, and compared to their actual administered doses. Accordingly, doses were categorized as rounded up, rounded down, not rounded at all, or missing (due to not being entered into the system or unavailability of a hard copy documenting the actual amount of the administered dose). Based on all the administered doses for each patient, the patients were categorized into one of the following four groups: (1) rounding up ± no rounding (at least one of the administered doses was higher than the calculated dose and other administered doses were equal to the calculated doses); (2) rounding down ± no rounding (at least one of the administered doses was lower than the calculated dose and other administered doses were equal to the calculated doses); (3) no rounding at all (all administered doses were equal to the calculated doses); and (4) mixed or missing (at least one of the administered doses was higher, at least one of the administered dose was lower, at least one of the administered doses was equal to the calculated dose, or at least one dose was missing).

### 2.3. Statistical Analysis

A Student’s *t*-test for continuous variables and a chi-squared test for categorical variables were used to assess the significance of differences in adult and pediatric patients. Correlations between neuropathy, baseline characteristics, and clinical parameters were estimated separately in adult and pediatric groups. The correlation between neuropathy and dose rounding was also estimated in adult as well as pediatric patients. All statistical analyses were performed using SPSS software version 28 (IBM Corp., Armonk, NY, USA); a *p*-value < 0.05 indicated significant results.

## 3. Results

### 3.1. Demographic and Clinical Characteristics of the Enrolled Subjects

A total of 524 patients were identified in the initial screening for this study. Of these, 141 were duplicates, 22 had administered vincristine before 2016, and 6 had undocumented clinical variables. This yielded 355 patients receiving vincristine for the first time between 2016 and 2022 to be included in the analysis of the frequency of neuropathy (Figure 1). Two patients had missing ages and, thus, were not included in determining the influence of dose rounding or clinical parameters on neuropathy.

Of the 355 patients, there were 147 pediatric and 206 adult patients with a mean age of 5.32 and 49 years, respectively (Table 1). The majority of pediatric patients were diagnosed with solid malignancies (71.9%), while the most significant portion of adult patients were diagnosed with hematologic/lymphatic malignancies (69.6%). Multiple comorbidities were noticed in >90% of adult patients. Further details about the demographic and clinical data are provided in Table 1.

### 3.2. Factors Affecting the Frequency of Vincristine-Induced Neuropathy

Regardless of the patient’s age, 34.6% of them developed neuropathic symptoms following vincristine use (Figure 2a), among which 55.3% reported autonomic symptoms of neuropathy, followed by peripheral (19.5%) and cranial (13.8%) neuropathies (Figure 2b). Interestingly, although there was no significant difference in the proportion of patients who suffered from autonomic neuropathy in both pediatric and adult patient populations (55.1% and 56.1%, respectively), cranial neuropathy seemed to be more frequent in pediatric patients (22.5%) (Figure 3b). In comparison, a higher rate of peripheral neuropathy was noted in adults (24.7%) (Figure 4b).

Further analyses were conducted to determine other factors that could affect the incidence of neuropathy in these patients. Table 2a shows that none of the factors, including patient gender, type of malignancy, presence or absence of different and multiple comorbidities, and use of azole antifungal medications, significantly impact neuropathic episodes in pediatric patients. On the contrary, the existence of respiratory comorbidities, such as asthma (*p* = 0.044), and the concurrent use of azole antifungals (*p* < 0.001) were significantly associated with an increase in the frequency of neuropathy in adult patients, as presented in Table 2b. Moreover, the absence of endocrinological comorbidities, such as diabetes and hypothyroidism, was significantly (*p* = 0.019) associated with the absence of neuropathic symptoms in adult patients (Table 2b). Regarding the association between neuropathy and clinical parameters, pediatric patients who showed symptoms of neuropathy had significantly higher BSA (*p* = 0.038), serum creatinine (Scr) (*p* = 0.044), and numerically higher levels of total bilirubin levels (*p* = 0.062), as depicted in Table 3a. No clinical parameter was associated with the presence or absence of neuropathic symptoms in adult patients, except for that of higher WBC counts in patients with reported neuropathy, which did not reach significance (*p* = 0.081; Table 3b).

### 3.3. Impact of Dose Rounding on the Incidence of Neuropathy

To investigate the association between vincristine dose rounding and the neuropathy frequency among adult and pediatric patients, the proportion of those with and without neuropathy in each dose-rounding group was estimated. Overall, the patients who received one or more rounded-up dose of vincristine reported more neuropathy symptoms (71.4% vs. 28.6%, *p* < 0.001). On the other hand, patients who received one or more rounded-down dose of vincristine had a reduced incidence of neuropathic pain (73.6% vs. 26.4%, *p* < 0.001) (Table 4a). The observed positive association between rounding-up doses and the presence of neuropathy was more prevalent in pediatric patients (76.2%, n = 16 of 21). In comparison, the association between rounding-down doses and the absence of neuropathy was predominant in adult patients (77.8%, n = 105 of 135) (Table 4b,c). A total of 33 patients (5 pediatrics and 28 adults) reported symptoms of neuropathy, even when all administered vincristine doses were equal to the calculated ones (no rounding at all). Moreover, despite the age, the majority of patients who either received mixed dosing or missed at least one dose did not show any neuropathy manifestations (*p* < 0.001; Table 4a–c).

## 4. Discussion

In this study, we sought to determine the frequency of neuropathy and investigate any association with dose rounding in patients diagnosed with solid, hematologic, and lymphatic malignancies who were administered, for the first time, vincristine-based chemotherapy. Approximately one-third (n = 123) of patients (73 of 206 adults, 49 of 147 children, and 1 patient with a missing age) developed neuropathy after vincristine administration. Regardless of age, most patients reported symptoms of an autonomic type of neuropathy, including hypotension, constipation, urinary retention, and incontinence, while the remaining patients suffered from peripheral, cranial, or a combination of multiple types of neuropathies.

### 4.1. Vincristine-Induced Neuropathy in Pediatric Patients

Mounting evidence suggests a strong association between vincristine use and the development of different types of neuropathies in pediatric patients at different rates. For instance, Wali Y. et al. reported that 18.4% of pediatric patients developed vincristine-related peripheral and autonomic neuropathies [11]. Furthermore, previous studies have demonstrated that 33.75% [21] and 29.7% [22] of children developed a mainly peripheral type of neuropathy following vincristine administration. Our result, 33.3% of pediatric patients suffered from neuropathy, falls into this range. More importantly, our results reveal autonomic neuropathy as the most common type that develops in pediatric patients, followed by the more serious form, cranial neuropathy, and then peripheral neuropathy, which resembles the results reported by Wali Y. et al. [11]. Overall, the type of neuropathy and proportion of patients developing vincristine-induced neuropathy might depend on the patient population, the cumulative administered dose, and the criteria and tools used to diagnose neuropathy [23]

Several factors were analyzed for their potential impact on the rate of neuropathy, including gender, type of malignancy, presence of different and multiple comorbidities, and the concurrent use of azole antifungals. None of those seemed to significantly affect the rate of reported neuropathy in pediatric patients (Table 2a). While some studies have reported higher rates of vincristine-induced neuropathy among pediatric males [21,24], our results are consistent with others where females had a higher, yet non-significant, rate of neuropathy [25,26]. Although the current evidence does not support the notion that patients with pre-existing neurologic conditions are at a higher risk of neuropathy with vincristine use [27], our study reveals a trend of more frequent vincristine-induced neuropathies in pediatric patients with pre-existing neurologic conditions (*p* = 0.077). When examining the clinical parameters, significantly higher BSA and creatinine serum levels were found in pediatric patients suffering from neuropathy compared to patients with no symptoms of neuropathy. The results also show that the presence of vincristine-induced neuropathy is associated with a statistically insignificant elevation of ALT (*p* = 0.135), AST (*p* = 0.235), and total bilirubin levels (*p* = 0.062; Table 3a). Tunjungsari DA et al. showed a significant association between elevated ALT and AST and a higher risk of neuropathy in pediatric patients [26]. However, the patients included in their study had impaired hepatic functions, while ours did not, as depicted in the range of liver enzymes (Table 3a). Our results show that vitamins B12 and D are numerically lower in pediatric patients with reported neuropathic symptoms. Such an observation reiterates what others have demonstrated: that pediatric patients with neuropathy have lower levels of both vitamins than those without neuropathy [28,29]. Thus, future studies are still warranted to determine the role of vitamins B12 and D as potential predictive biomarkers of vincristine-induced neuropathy and investigate the impact of their supplementation as a preventative strategy against neuropathy associated with vincristine use.

### 4.2. Vincristine-Induced Neuropathy in Adult Patients

Several lines of evidence have reported that the incidence of neuropathy, mainly the peripheral type, is approximately 30–40% in adult patients treated with vincristine [30,31]. Furthermore, the proportion of vincristine-induced neuropathy can reach as high as 70%, particularly among lymphoma patients [32]. Our result falls into this range, with 35.4% of adult patients treated with vincristine for different cancers having neuropathy (Figure 4a). Similar to pediatric patients, the autonomic type of neuropathy was the most prevalent type, followed by peripheral neuropathy. Our findings support what has been reported previously by Imam et al., where 80% of Sudanese patients developed autonomic neuropathic symptoms following vincristine treatment, while peripheral neuropathy was reported as the second most common neuropathic change in their study population [33].

Our data show that the absence of pre-existing respiratory and endocrinological diseases is significantly associated with lower rates of vincristine-induced neuropathy in adult patients (*p* < 0.05; Table 2b). The triazole antifungal agents posaconazole, voriconazole, fluconazole, and itraconazole are commonly used for prophylaxis and treating invasive fungal infections during invasive chemotherapy. They interact with the enzymes responsible for vincristine metabolism and thus potentiate the latter’s side effects [16]. Our results show that the co-administration of azole antifungals with vincristine results in a significantly higher frequency of neuropathy in adult patients (*p* < 0.001; Table 2b). Interestingly, most of these patients received fluconazole. Even though a higher incidence and severe neuropathy were reported upon treatment with other members of the azole family [7], our results agree with others in that the judicious use of azole antifungals is warranted in vincristine-treated patients, as the risk of neuropathy remains with fluconazole, even when it is used at the therapeutic dose [34]. With respect to the clinical parameters, an elevated level of WBCs was found in adults with neuropathy, which was not observed in pediatric patients. Such an elevation can be attributed to the higher incidence of fungal infections in adults, and hence, increased use of azole antifungals compared to pediatric patients. This might perpetuate the significant increase in the frequency of neuropathy associated with azole antifungals observed among adults compared to pediatric patients. Neither hepatic function, renal function, nor BMI significantly impacted the frequency of neuropathy cases, which resembles the results reported by Okada et al. [35].

### 4.3. Vincristine Dose Rounding and Neuropathy

The dose rounding of cytotoxic agents has been advocated in oncology as a cost-saving strategy for streamlining workloads and facilitating drug administration [18,19,20]. Despite the mounting evidence demonstrating its cost-effectiveness [36,37,38], there is a lack of studies regarding its impact on the occurrence of side effects, particularly vincristine-induced neuropathy. Intriguingly, investigating this practice and its impact on the safety and effectiveness of antineoplastic therapies is encouraged by the Hematology/Oncology Pharmacy Association (HOPA) [20]. Our findings show a significant association between the dose rounding of vincristine and the proportion of patients diagnosed with neuropathy after their first-time exposure to vincristine. These findings should be interpreted cautiously, as several previous studies showed dose rounding within ±5% of the calculated chemotherapy dose to be clinically tolerated without yielding any adverse effects [36,39]. Several confounders may explain the conflicting results observed in this study compared to previous studies, including the investigated cytotoxic agent, other drugs included in the treatment protocol, previous exposure to the cytotoxic drug or other drugs with similar adverse effect profiles, the purpose of the treatment regimen (palliative vs. curative), and more importantly, the magnitude of dose rounding. Thus, considering all the factors mentioned earlier, the patient criteria must be recognized when studying such an issue. Cumulatively, given the scarcity of the research on this topic, further research is warranted in the near future.

## 5. Implications and Limitations

This study identified the factors that can affect vincristine-induced neuropathy and the effect of dose rounding on the frequency of neuropathy occurrence in pediatric and adult patients in Saudi Arabia. The findings from our study may help clinicians and decision makers revise vincristine-containing treatment protocols and interventions to avoid the potential development of adverse neuropathy effects while maintaining effectiveness. Nevertheless, our study results should be interpreted cautiously to avoid imprecise generalization to other cases where dose rounding is implemented.

Several limitations and challenges exist in our study that are worth highlighting. First, the rate of neuropathy was based on the patient’s complaint of neuropathy symptoms solely, and no tests were performed to confirm them. Unfortunately, given the retrospective nature of this study, we had no control over applying such tests. Although this could potentially impact the rate of existing neuropathy, we noticed that no clinical tests were performed, even when patients complained of foot drop or tingling/numbness/burning in their extremities after vincristine administration, which certainly indicated the presence of neuropathy. Despite using a retrospective study design, we believe that using different tools, such as questionnaires directed towards patients, or using a different study design, such as a prospective study, in addition to tests confirming neuropathy occurrence capture the exact rate better.

Second, other factors that may have interfered with vincristine metabolism directly or indirectly, such as using neurokinin inhibitors to mitigate chemotherapy-induced nausea and vomiting, were not determined. Although such data are critical, the main focus of our study was the impact of vincristine dose rounding on the frequency of neuropathy.

Third, we did not determine the association between the total amount and number of administered doses and the occurrence of neuropathy. This sounds important; however, the considerable variation in the number and number of doses, even among those who developed neuropathy, would lead to an overlap with those who had no neuropathy and thus, conceal any associated significance.

Fourth, the dose rounding of the vincristine dose may also have an influence on tumor response. Although this is clinically important, this seems impossible to demonstrate in our study due to the small number of patients and lack of following-up data for our patients.

Despite these limitations, our study is the first of its kind in the oncology field, and all limitations will be considered in the upcoming studies. Our future direction would be to differentiate between early onset vs. late-onset neuropathy among those who were administered vincristine for the first time. Furthermore, we will determine the range to which we can round doses up without developing side effects, including neuropathy, and round down without impacting treatment effectiveness.

## 6. Conclusions

Our study demonstrated that autonomic neuropathy was the most common type of vincristine-related neuropathy. A more serious type of neuropathy, cranial neuropathy, was observed more frequently among pediatric patients. Dose rounding was a significant determinant of neuropathy development in pediatric and adult patients. The concurrent use of azole antifungals notably enhanced vincristine-mediated neuropathy in adult patients. In light of these results, studies are still needed to continue evaluating the confounders and preventative measures that can exacerbate or prevent the development of vincristine-induced neuropathy, respectively. Furthermore, more studies are still warranted to assess the safe yet effective range of vincristine dose rounding for first-time users.

## Figures and Tables

**Figure 1 jcm-12-05662-f001:**
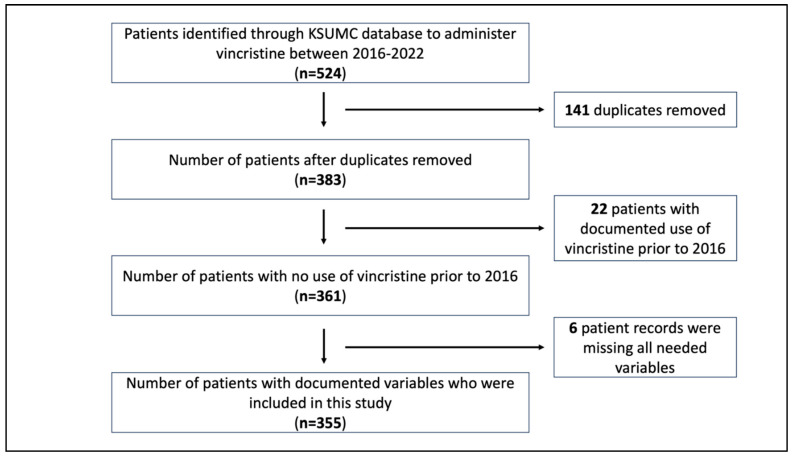
Flowchart demonstrating criteria of patients admitted to the study.

**Figure 2 jcm-12-05662-f002:**
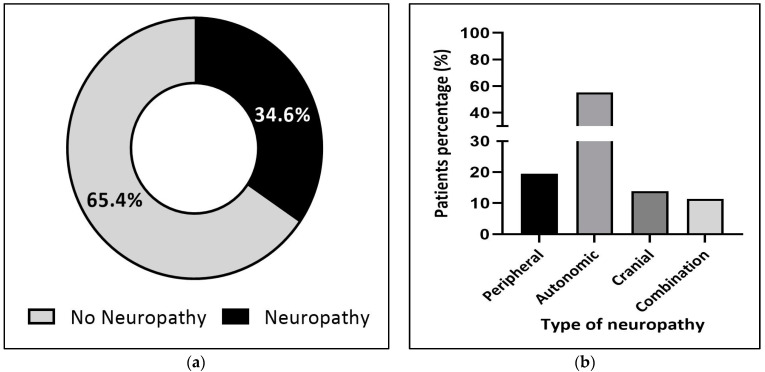
(**a**) Proportion of neuropathy in all patients (n = 355). (**b**) Classification of all patients according to the type of neuropathy (n = 123). Types of neuropathy: **peripheral** (loss of deep tendon reflexes, paresthesia, neurotic pain, numbness, and wrist or foot drop); **autonomic** (constipation, bladder atony with retention of urine, and abdominal pain); **cranial** (transient blindness, diplopia, seizures, confusion, aphasia, jaw pain, facial palsy, hearing loss, and vocal cord paresis or paralysis); **combination** (2 or 3 types).

**Figure 3 jcm-12-05662-f003:**
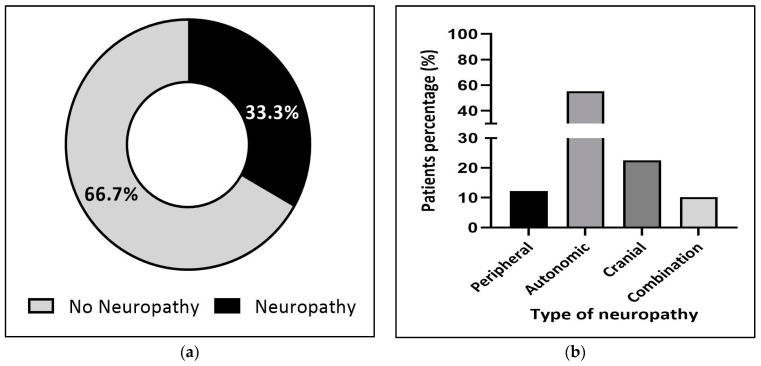
(**a**) Proportion of neuropathy in pediatric patients (n = 147). (**b**) Pediatric patients’ classification according to the type of neuropathy (n = 49).

**Figure 4 jcm-12-05662-f004:**
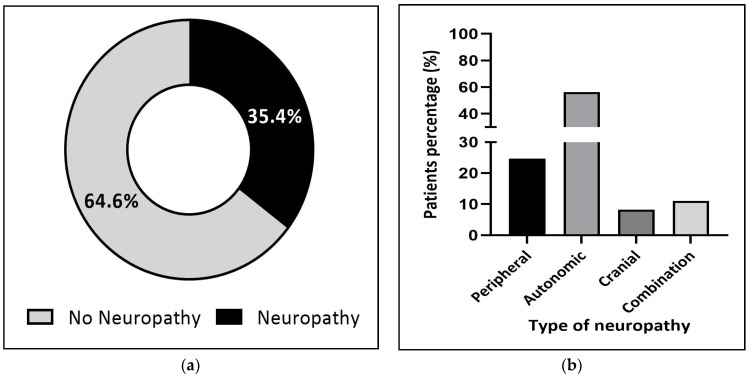
(**a**) Proportion of neuropathy in adult patients (n = 206). (**b**) Adult patients’ classification according to the type of neuropathy (n = 73).

**Table 1 jcm-12-05662-t001:** Baseline characteristics of patients (n = 355).

Characteristics	All (n = 355)	Pediatric (n = 147)	Adult (n = 206)	*p*-Value
	**n (Frequency %)**			
**Gender**				0.64
Male	215 (60.6)	87 (40.7)	127 (59.3)	
Female	140 (39.4)	60 (43.2)	79 (56.8)	
**Malignancies**				**0.000**
Solid	96 (27.0)	69 (71.9)	27 (28.1)	
Hematologic/lymphatic	259 (73.0)	78 (30.4)	179 (69.6)	
**Comorbidities ***	130 (36.6)	23 (17.8)	106 (82.2)	**0.000**
**Comorbidities ***				
Neurological	18 (5.1)	6 (33.3)	12 (66.7)	0.463
Cardiovascular	72 (20.3)	7 (9.7)	65 (90.3)	**0.000**
Respiratory	15 (4.2)	4 (26.7)	11 (73.3)	0.229
Endocrinological	56 (15.8)	5 (8.9)	51 (91.1)	**0.000**
Renal	17 (4.8)	1 (5.9)	16 (94.1)	**0.002**
Other	48 (13.5)	6 (12.8)	41 (87.2)	**0.000**
**Total Comorbidities**				**0.000**
0	225 (63.5)	124 (55.4)	100 (44.6)	
1	58 (16.3)	18 (31.6)	39 (68.4)	
2	52 (14.6)	4 (7.7)	48 (92.3)	
3	16 (4.5)	1 (6.3)	15 (93.8)	
4	4 (1.1)	-	4 (100)	
**Parameters [Mean** **± SD]**				
Age (years) *	30.83 ± 26.6	5.32 ± 3.9	49.02 ± 20.1	**0.000**
BMI *	22.88 ± 7.7	16.68 ± 4.2	27.29 ± 6.5	**0.000**
BSA *	1.39 ± 0.6	0.80 ± 0.4	1.81 ± 0.3	**0.000**
Hgb (g/L) *	109.69 ± 23.8	104.22 ± 20.3	113.36 ± 25.4	**0.000**
RBC (×10^9^/L) *	4.47 ± 5.7	4.11 ± 0.9	4.73 ± 7.4	0.326
MCV (fL) *	78.98 ± 12.3	76.60 ± 13.4	80.57 ± 11.3	**0.003**
WBC (×10^9^/L) *	14.15 ± 34.3	16.41 ± 42.9	12.59 ± 26.6	0.306
Neutrophil count (×10^9^/L) *	5.03 ± 4.6	4.71 ± 5.6	5.21 ± 3.9	0.337
Lymphocyte count (×10^9^/L) *	4.14 ± 15.4	3.69 ± 3.6	4.47 ± 19.6	0.588
NLR *	4.54 ± 8.2	3.46 ± 9.4	5.19 ± 7.34	**0.064**
Vit. D (nmol/L) *	68.97 ± 75.1	68.80 ± 67.8	69.02 ± 77.3	0.993
Vit. B12 *	505.61 ± 459.2	350.2 ± 185.2	534.40 ± 481.4	0.161
ALT (µ/L) *	39.40 ± 84.2	31.56 ± 25.9	44.90 ± 106.9	0.153
AST (µ/L) *	38.56 ± 45.3	37.92 ± 27.4	38.95 ± 54.2	0.821
Total Bilirubin (µmol/L) *	11.28 ± 26.5	6.53 ± 7.0	14.54 ± 33.6	**0.001**
Scr (µmol/L) *	62.03 ± 62.5	35.45 ± 26.5	81.16 ± 73.2	**0.000**

* Missing information from patients in each group.

**Table 2 jcm-12-05662-t002:** (**a**) Baseline characteristics influencing or tending to influence neuropathy occurrence in pediatric patients (n = 147) using chi-squared test. (**b**) Baseline characteristics influencing or tending to influence neuropathy occurrence in adult patients (n = 206) using chi-squared test.

Characteristics	Variable/Occurrence	Neuropathy		*p*-Value
		Absent n (%)	Present n (%)	
(**a**)				
**Gender**	M	60 (69.0)	27 (31.0)	0.476
	F	38 (63.3)	22 (36.7)	
**Malignancy**	Solid	49 (71.0)	20 (20.0)	0.293
	Hematologic/lymphatic	49 (62.8)	29 (37.2)	
**Comorbidity**	No	85 (68.5)	39 (31.5)	0.261
	Yes	13 (56.5)	10 (43.5)	
**Comorbidities**				
Neurological	No	96 **(68.1)**	45 (31.9)	**0.077**
	Yes	2 (33.3)	4 **(66.7)**	
Cardiovascular	No	93 (66.4)	47 (33.6)	0.784
	Yes	5 (66.7)	2 (28.6)	
Respiratory	No	96 (67.1)	47 (32.9)	0.473
	Yes	2 (50.0)	2 (50.0)	
Endocrinological	No	95 (66.9)	47 (33.1)	0.748
	Yes	3 (60.0)	2 (40.0)	
Renal	No	97 (66.4)	49 (33.6)	0.478
	Yes	1 (100.0)	0 (0.0)	
Others	No	94 (66.7)	47 (33.3)	1.00
	Yes	4 (66.7)	2 (33.3)	
**Total comorbidity**	0	85 (68.5)	39 (31.5)	0.532
	1	10 (55.6)	8 (44.4)	
	2	2 (50.0)	2 (50.0)	
	3	1 (100.0)	0 (0.00)	
**Concurrent use of azole Antifungal**	No	97 (67.4)	47 (32.6)	0.216
	Yes	1 (33.3)	2 (66.7)	
(**b**)				
**Gender**	M	80 (63.0)	47 (37.0)	0.550
	F	53 (67.1)	26 (32.9)	
**Malignancy**	Solid	16 (59.3)	11 (40.7)	0.537
	Hematologic/lymphatic	117 (65.4)	62 (34.6)	
**Comorbidity**	No	69 (69.0)	31 (31.0)	0.196
	Yes	64 (60.4)	42 (39.6)	
**Comorbidities**				
Neurological	No	124 (63.9)	70 (36.1)	0.436
	Yes	9 (75.0)	4 (25.0)	
Cardiovascular	No	94 (66.7)	47 (33.3)	0.353
	Yes	39 (60.0)	26 (40.0)	
Respiratory	No	129 **(66.2)**	66 (33.8)	**0.044**
	Yes	4 (36.4)	7 **(63.6)**	
Endocrinological	No	107 **(69.0)**	48 (31.0)	**0.019**
	Yes	26 (51.0)	25 (49.0)	
Renal	No	121 (63.7)	69 (36.3)	0.363
	Yes	12 (75.0)	4 (25.0)	
Others	No	111 (67.3)	54 (32.7)	0.103
	Yes	22 (53.7)	19 (46.3)	
**Total comorbidity**	0	69 (69.0)	31 (31.0)	0.253
	1	26 (66.7)	13 (33.3)	
	2	30 (62.5)	18 (37.5)	
	3	6 (40.0)	9 (60.0)	
	4	2 (50.0)	2 (50.0)	
**Concurrent use of azole Antifungal**	No	128 **(69.6)**	56 (30.4)	**<0.001**
	Yes	5 (22.7)	17 **(77.3)**	

**Table 3 jcm-12-05662-t003:** (**a**) Clinical parameters associated with or tending to associate with neuropathy occurrence in pediatric patients (n = 147) using chi-squared test. (**b**) Clinical parameters associated with or tending to associate with neuropathy occurrence in adult patients (n = 206) using chi-squared test.

Parameter	Neuropathy		*p*-Value
	Absent	Present	
	Mean ± SD (n)		
(**a**)			
BMI	16.4 ± 3.7 (98)	17.3 ± 5.2 (49)	0.220
BSA	**0.76** ± 0.3 (98)	**0.89** ± 0.4 (49)	**0.038**
Hgb (g/L) *	102.7 ± 21.8 (97)	107.3 ± 16.5 (49)	0.199
RBC (×10^9^/L) *	4.11 ± 0.9 (96)	4.1 ± 0.9 (48)	0.84
MCV (fL) *	76.60 ± 13.8 (96)	76.6 ± 12.6 (48)	0.99
WBC (×10^9^/L) *	16.8 ± 48.1 (97)	15.6 ± 30.8 (49)	0.875
Neutrophil count (×10^9^/L) *	4.7 ± 5.6 (85)	4.8 ± 5.5 (43)	0.928
Lymphocyte count (×10^9^/L) *	3.4 ± 2.6 (87)	4.3 ± 4.9 (43)	0.216
NLR *	3.8 ± 11.1 (85)	2.7 ± 4.4 (43)	0.533
Vit. D (nmol/L) *	79.3 ± 83.1 (7)	50.4 ± 28.5 (4)	0.525
Vit.B12 *	387.5 ± 190.9 (4)	201.0 (1)	0.447
ALT (µ/L) *	28.8 ± 20.4 (91)	36.9 ± 33.9 (47)	0.135
AST (µ/L) *	35.6 ± 22.9 (89)	42.3 ± 34.2 (47)	0.235
Total Bilirubin (µmol/L) *	**5.5** ± 3.8 (92)	**8.6** ± 10.7 (46)	**0.062**
Scr (µmol/L) *	**32.2** ± 18.8 (93)	**41.6** ± 36.5 (49)	**0.044**
(**b**)			
BMI *	27.6 ± 6.7 (132)	26.7 ± 6.0 (73)	0.324
BSA *	1.8 ± 0.3 (132)	1.8 ± 0.2 (73)	0.291
Hgb (g/L)	115.4 ± 23.8 (133)	109.7 ± 27.9 (73)	0.128
RBC (×10^9^/L) *	4.22 ± 0.9 (133)	5.7 ± 12.5 (72)	0.182
MCV (fL) *	81.2 ± 11.9 (133)	79.5 ± 10.1 (72)	0.324
WBC (×10^9^/L) *	**9.6** ± 14.68 (133)	**18.1** ± 39.75 (73)	**0.081**
Neutrophil count (×10^9^/L) *	4.9 ± 3.5 (129)	5.8 ± 4.6 (69)	0.110
Lymphocyte count (×10^9^/L) *	2.9 ± 8.5 (128)	7.3 ± 30.9 (70)	0.136
NLR *	5.4 ± 8.00 (128)	4.8 ± 6.1 (69)	0.611
Vit. D (nmol/L) *	62.7 ± 83.5 (20)	80.6 ± 70.8 (11)	0.551
Vit.B12 *	403.9 ± 395.7 (14)	674.9 ± 557.3 (13)	0.155
ALT (µ/L) *	46.5 ± 129.7 (129)	42.1 ± 46.5 (73)	0.782
AST (µ/L) *	39.6 ± 59.2 (128)	37.8 ± 44.9 (73)	0.818
Total Bilirubin (µmol/L) *	16.5 ± 41.0 (131)	11.0 ± 10.4 (72)	0.260
Scr (µmol/L) *	84.4 ± 86.5 (124)	75.4 ± 40.4 (69)	0.419

* Missing information from patients in each group.

**Table 4 jcm-12-05662-t004:** (**a**) Dose rounding of vincristine and neuropathy observed in all patients using chi-squared test (n = 355). (**b**) Dose rounding of vincristine and neuropathy occurrence in pediatric patients using chi-squared test (n = 147). (**c**) Dose rounding of vincristine and neuropathy occurrence in adult patients using chi-squared test (n = 206).

Nature of Dose Rounding	n (%)	Neuropathy		*p*-Value
		Absent n (%)	Present n (%)	
(**a**)				
Rounding up ± no rounding	28 (7.8)	8 (28.6)	20 **(71.4)**	**<0.001**
Rounding down ± no rounding	193 (54.3)	142 **(73.6)**	51 (26.4)	
No rounding at all	34 (9.5)	1 (2.9)	33 **(97.1)**	
Mixed or missing	100 (28.1)	81 **(81.0)**	19 (19.0)	
(**b**)				
Rounding up ± no rounding	21 (14.2)	5 (23.8)	16 **(76.2)**	**<0.001**
Rounding down ± no rounding	57 (38.7)	36 **(63.2)**	21 (36.8)	
No rounding at all	6 (4.0)	1 (16.7)	5 **(83.3)**	
Mixed or missing	63 (42.8)	56 (**88.9)**	7 (11.1)	
(**c**)				
Rounding up ± no rounding	7 (3.4)	3 (42.9)	4 **(57.1)**	**<0.001**
Rounding down ± no rounding	135 (65.5)	105 **(77.8)**	30 (22.2)	
No rounding at all	28 (13.6)	0 (0.0)	28 **(100)**	
Mixed or missing	36 (17.5)	25 **(69.4)**	11 (30.6)	

## Data Availability

Data presented in this study are available upon request from the corresponding author. Data are not publicly available due to privacy reasons.

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
