# Peer review of "Vincristine-Induced Neuropathy in Patients Diagnosed with Solid and Hematological Malignancies: The Role of Dose Rounding"

_jcm, 2023, doi:10.3390/jcm12175662_

Round 1

Reviewer 1 Report

The paper analyzes the role of Dose Rounding of Vincristine and the induced neuropathy in patients diagnosed with solid and hematological malignancies.

Unfortunately no table sand figures are included in the downloaded manuscript. Therefore some questions may not be able to answer. Nevertheless there are some open question in the result section:

1. In adults how many patients did recieve the the single maximum dose of 2 mg and were below the the calcculated dose by BSA? This may have influenced the results, as there must be many adults than children receiving the maximum single dose.

2. Is neurotoxicity also depending on the cumulative dose of vincristine or the number of applications? Is neurotoxicity also depending on the number of vincristine applications and the time interval between following vincristine applications? Lines 361 to 365 are not convincing to do this analysis.

4. When after after treatment start of vincristine or how many doses of vincristine did the neurotoxicity occur? Is there a correlation between these factors and neuirotoxicity?

5. In how many patients was vincristin stopped and not given after occurence of neurotoxicity?

Discussion: In the discussion several new results are presented like the association with vitamins (line 283 ff) that are not mentioned in the results. ;Maybe this is given in the not available tables.

T he rounding of the vincristine dose may also have an influence on tumor response. This should be mentioned in the discussion. It seems unlikely to demonstrate this by the small number of patients for the different diseases. Nevertheless this is worthwhile to mention.

Minor points:

line 145: delete the word "the" before clinical

line 249: 60% of 206 adults and 40% of 147 children are more than 123. Numbers need to be checked.

Author Response

We would like to thank the reviewer for their time in commenting on the manuscript. Below is point-by-point response to their comments.  

Reviewer 1:

The paper analyzes the role of Dose Rounding of Vincristine and the induced neuropathy in patients diagnosed with solid and hematological malignancies. Unfortunately, no table sand figures are included in the downloaded manuscript. Therefore, some questions may not be able to answer. Nevertheless, there are some open question in the result section

Response: We are truly sorry about that. Tables and figures are now attached in this version of the manuscript.

In adults how many patients did receive the single maximum dose of 2 mg and were below the calculated dose by BSA? This may have influenced the results, as there must be many adults than children receiving the maximum single dose.

Response: We appreciate the reviewer for this valid comment. During data collection, we have not documented the maximum doses administered to each patient regardless of age. Although this is truly important, our data collection approach should have considered that. Our approach, however, was slightly different: to calculate the dose received based on BSA at each time, compare it to the actual administered dose, determine the type of rounding, then calculate the cumulative amount that was rounded up or down. All of this was done simultaneously without documenting all details for each dose. Sadly, given the main focus of the study and several steps to calculate dose rounding, we will not be able to answer this question. However, after looking at the number of patients’ doses rounded up in adults and pediatric patients, none of them have all doses rounded up only; hence we had different classifications considering each cohort separately, as shown in Tables 4 b and c.

Is neurotoxicity also depending on the cumulative dose of vincristine or the number of applications? Is neurotoxicity also depending on the number of vincristine applications and the time interval between following vincristine applications? Lines 361 to 365 are not convincing to do this analysis.

Response: We appreciate the reviewer for this comment. We believe that neuropathy associated with vincristine depends on the cumulative dose and number of applications. Although cumulative dose has been discussed previously (PMID: 3540519), we could not investigate that in our study, and we openly considered that as a limitation of our study. The interval between applications sounds intriguing; however, its impact will not be significant, especially since it is fixed among most patients administered vincristine according to the used protocols.

When after treatment start of vincristine or how many doses of vincristine did the neurotoxicity occur? Is there a correlation between these factors and neurotoxicity?

Response: We appreciate the reviewer for this comment. Neurotoxicity occurred after an average of 6.5 doses in pediatric patients and 3 doses in adults. This reiterates the notion that cumulative dose of vincristine plays a major role in its occurrence as higher doses based on BSA are expected to be given to adult patients. Despite that, we could not investigate it for the huge variation in number, hence amount of doses and we highlighted that as limitation of our study. We believe the only way to reveal such a relation is to conduct this study prospectively and on population matched in their characteristics.

In how many patients was vincristine stopped and not given after occurrence of neurotoxicity?

Response: We appreciate the reviewer for this comment. In fact, the idea of this project was initially to assess the interventions done towards neuropathy associated with vincristine use. Due to lack of information about the incidence of neuropathy in our institution, we had to conduct this study. Although the rate was 33% to 35%, only 9 adult patients (12.3%) and 8 pediatric patients (16.3%) were intervened. Interventions were either reducing the dose (4 in adult, 1 in pediatrics), holding the dose (2 adults and 7 pediatrics) or combination (3 adults only). Results were in fact shocking!!

Discussion: In the discussion several new results are presented like the association with vitamins (line 283 ff) that are not mentioned in the results. ; Maybe this is given in the not available tables.

Response: We appreciate the reviewer for this comment. Since Tables are now included, this should clarify the association of neuropathy with vitamins.

The rounding of the vincristine dose may also have an influence on tumor response. This should be mentioned in the discussion. It seems unlikely to demonstrate this by the small number of patients for the different diseases. Nevertheless, this is worthwhile to mention.

Response: We appreciate the reviewer for this comment. We have clearly stated that as a limitation of our study (line 369-371).

Minor points:

line 145: delete the word "the" before clinical

Response: We appreciate the reviewer for this comment. “The” was deleted as requested.

line 249: 60% of 206 adults and 40% of 147 children are more than 123. Numbers need to be checked.

Response: We apologize for this confusion. A total number of those with neuropathy in our study is 123 out of 355. Of those 123, 73 were adults and 49 were pediatric, which represent 59.4% and 39.8%, respectively. The age of one patient is missing. To avoid this confusion, we have changed this sentence to “Approximately one-third (n=123) of patients (73 of 206 adults, 49 of 147 children, one patient with missing age) have developed neuropathy after vincristine administration”. (Lines 251-252)

Reviewer 2 Report

The manuscript clearly describes the frequency and risk factors of VCR-induced neuropathy in pediatric and adults population. 

The limitation of the study is it's retrospective character which unables the objective diagnosis of neuropathy, but the authors write about it in the manuscript.

I realize that it may be difficult in the retrospective analysis, but if possible, it would be worth to add

1. if there were any patients with HIV infection which may increase the risk of neuropathy

2. Why the azoles were used in patients, althought they are contrindicated if VCR is given? please add at least a comment

3. How many patients had the dose rounded of more than 5%?

No comments - I am not a native speaker, but the manuscript reads well.

Author Response

We would like to thank the reviewer for their time in commenting on the manuscript. Below is point-by-point response to their comments.  

Reviewer 2:

The manuscript clearly describes the frequency and risk factors of VCR-induced neuropathy in pediatric and adults population. 

The limitation of the study is it's retrospective character which unables the objective diagnosis of neuropathy, but the authors write about it in the manuscript.

I realize that it may be difficult in the retrospective analysis, but if possible, it would be worth to add

Response: We appreciate the reviewer for his/her understanding of the difficulty in conducting such study with the limited information we have.

If there were any patients with HIV infection which may increase the risk of neuropathy

Response: We appreciate the reviewer for the comment. There is only 1 adult patient with HIV and he developed neuropathy. Unfortunately, there is no enough number in the study sample to investigate this correlation.

Why the azoles were used in patients, although they are contraindicated if VCR is given? please add at least a comment.

Response: We appreciate the reviewer for the comment. Despite their ability to interrupt with VCR metabolism, it is universally accepted that vincristine-treated patients who are having fungal infection or at high risk of developing fungal infection are candidate for azoles treatment as they are the preferred class for that.

How many patients had the dose rounded of more than 5%?

Response: We appreciate the reviewer for this valid comment. During data collection, we have not documented the number of doses rounded >5% to each patient regardless of age. Although this is truly important, our data collection approach, unfortunately, did not consider that. Our approach, however, was slightly different: to calculate the dose received based on BSA at each time, compare it to the actual administered dose, determine the type of rounding, then calculate the cumulative amount that was rounded up or down. All of this was done simultaneously without documenting all details for each dose. Sadly, we cannot answer this question in this study at this point, and we apologize for that.

No comments - I am not a native speaker, but the manuscript reads well.

Response: We appreciate the reviewer for this comment.

Reviewer 3 Report

Summary:

The manuscript titled "Vincristine-induced Neuropathy in Patients diagnosed with Solid and Hematological Malignancies: Role of Dose Rounding" addresses an interesting and relevant topic. However, there are several shortcomings that need to be addressed. The study focuses on investigating the role of dose rounding in vincristine-induced neuropathy among patients with various malignancies. While the subject is noteworthy, the manuscript lacks critical details in the Method and Results sections. The reviewer provides specific comments and suggestions for improvement in these areas.

General Comments:

The manuscript explores an important issue related to vincristine-induced neuropathy and its potential link to dose rounding strategies. However, the lack of methodological details, absence of essential patient characteristics, missing figures, and inadequate data presentation hinder the comprehensibility and scientific rigor of the study. Addressing these concerns is crucial for the manuscript's advancement.

Specific Comments:

A) Method Section:

The manuscript should explicitly state that appropriate ethical approval and informed consent, whether written or oral, were obtained to include patients' data in the study. Ethical considerations and patient privacy are paramount.

Detailed information about the data collection process is necessary, including qualifications of individuals involved in data collection and any mechanisms in place for data validation or double-checking.

The criteria for diagnosing each type of neuropathy should be clearly outlined. This will provide a foundation for readers to understand the clinical aspects of the study accurately.

If patients with pre-existing neurological symptoms were included, the manuscript must clarify the extent to which the chemotherapy regimen exacerbated their symptoms. The distinction between chemotherapy-related symptoms and pre-existing conditions is important for accurate interpretation.

Clearly define the inclusion and exclusion criteria for patient selection. This will help readers understand the characteristics of the patient population under investigation.

Incorporate a PRISMA diagram to provide a visual representation of the data collection, inclusion criteria, and exclusion criteria. This will enhance the transparency of the study methodology.

B) Results Section:

Present data in appropriate tables and figures to aid in clear data representation and to facilitate statistical analysis. Ensure that statistical analyses are appropriately conducted and that significant differences are indicated.

The absence of figures cited within the manuscript is concerning. All figures should be included in the PDF file to allow for proper evaluation by reviewers.

Regards,

17/8/2023

It is standard.

Author Response

We would like to thank the reviewer for their time in commenting on the manuscript. Below is point-by-point response to their comments.  

Reviewer 3:

The manuscript titled "Vincristine-induced Neuropathy in Patients diagnosed with Solid and Hematological Malignancies: Role of Dose Rounding" addresses an interesting and relevant topic. However, there are several shortcomings that need to be addressed.

The study focuses on investigating the role of dose rounding in vincristine-induced neuropathy among patients with various malignancies. While the subject is noteworthy, the manuscript lacks critical details in the Method and Results sections. The reviewer provides specific comments and suggestions for improvement in these areas.

Response: We appreciate the reviewer for this comment. We have addressed the comment raised by other reviewers regarding this part.

General Comments:

The manuscript explores an important issue related to vincristine-induced neuropathy and its potential link to dose rounding strategies. However, the lack of methodological details, absence of essential patient characteristics, missing figures, and inadequate data presentation hinder the comprehensibility and scientific rigor of the study. Addressing these concerns is crucial for the manuscript's advancement.

Response: We appreciate the reviewer for this comment. We have written the method that was followed exactly during data collection. Regarding the essential patient characteristics, we have collected what is clinically related to neuropathy, event NLR which was correlated with diabetic neuropathy (PMID: 29137012). We apologize for missing figures and tables. There are added to the revised version.

Specific Comments:

Method Section:

The manuscript should explicitly state that appropriate ethical approval and informed consent, whether written or oral, were obtained to include patients' data in the study. Ethical considerations and patient privacy are paramount.

Response: We appreciate the reviewer for this comment. Please refer to line 398-402

Detailed information about the data collection process is necessary, including qualifications of individuals involved in data collection and any mechanisms in place for data validation or double-checking.

Response: We appreciate the reviewer for this comment. Data collection was planned and lead by Dr. Abdulrahman A. Alwhaibi and Dr. Miteb A. Alenazi and frequent meeting were held to monitor and ensure consistency of data collection. For qualification, please refer to page 2 and 3.

The criteria for diagnosing each type of neuropathy should be clearly outlined. This will provide a foundation for readers to understand the clinical aspects of the study accurately.

Response: We appreciate the reviewer for this comment. We have stated this as a limitation for our study. Please refer to line 350-359.

If patients with pre-existing neurological symptoms were included, the manuscript must clarify the extent to which the chemotherapy regimen exacerbated their symptoms. The distinction between chemotherapy-related symptoms and pre-existing conditions is important for accurate interpretation.

Response: We appreciate the reviewer for this comment. We in fact collected the pre-existing neurological comorbidities (such as seizure and facial palsy) to investigate any potential correlation with the occurrence of neuropathy. However, there was nothing although results tended to show that in pediatric patients. Although the extent to which chemotherapy regimen exacerbates pre-existing neurological symptoms is clinically important, it is out of the scope of our study.

Clearly define the inclusion and exclusion criteria for patient selection. This will help readers understand the characteristics of the patient population under investigation.

Response: We appreciate the reviewer for this comment. We explained in the method section. We basically included all patients admitted to the oncology center at King Saud University Medical City (KSUMC) between 2016 and 2022 and received, for the first time, at least one dose of a vincristine-based chemotherapy regimen. Flow chart (Figure 1) has been developed to clarify that.

Incorporate a PRISMA diagram to provide a visual representation of the data collection, inclusion criteria, and exclusion criteria. This will enhance the transparency of the study methodology.

Response: We appreciate the reviewer for this comment. Flow chart (Figure 1) has been developed to clarify that.

Results Section:

Present data in appropriate tables and figures to aid in clear data representation and to facilitate statistical analysis. Ensure that statistical analyses are appropriately conducted and that significant differences are indicated. The absence of figures cited within the manuscript is concerning. All figures should be included in the PDF file to allow for proper evaluation by reviewers.

Response: We apologize for absence of figures and tables upon submitting this manuscript. All tables and figures are attached in the revised version. 

Reviewer 4 Report

Dear Authors, the research idea was original, the study was good and the conclusions were clear! However, I consider that you could somehow rephrase/improve on the abstract of the paper, as I consider that it does not emphasise well enough all the things that were studies, nor the importance of the findings(it does not catch the reader's attention from the first to the last sentence). Moreover, a moderate English correction is needed, more so in which concerns the spelling of various words. Respectfully yours, A Reviewer

Moderate English corrections are much needed.

Author Response

We would like to thank the reviewer for their time in commenting on the manuscript. Below is point-by-point response to their comments.  

Reviewer 4:

Authors, the research idea was original, the study was good and the conclusions were clear! However, I consider that you could somehow rephrase/improve on the abstract of the paper, as I consider that it does not emphasize well enough all the things that were studies, nor the importance of the findings (it does not catch the reader's attention from the first to the last sentence). Moreover, a moderate English correction is needed, more so in which concerns the spelling of various words.

Response: We really appreciate the reviewer for this comment. We have used MDPI language editing service to improve the language of our manuscript.

Round 2

Reviewer 1 Report

This is a revised version of the paper on the role of Dose Rounding of Vincristine and the induced neuropathy in patients diagnosed with solid and hematological malignancies.

According to the limitations of the study all comments to the reviewer are sufficiently answered.

Reviewer 3 Report

It can be accepted.

Without considerable grammar or syntax error.